# Catchment prioritization for freshwater mussel conservation in the Northeastern United States based on distribution modelling

Rebecca S. M. O'Brien[1]*, Graziella V. DiRenzo[2], Allison H. Roy[2], Jason Carmignani[3], Rebecca M. Quinones[3], Jennifer B. Rogers[1], Beth I. Swartz[4]

**1** Department of Environmental Conservation, Massachusetts Cooperative Fish and Wildlife Research Unit, University of Massachusetts, Amherst, Massachusetts, United States of America, **2** Department of Environmental Conservation, United States of America Geological Survey, Massachusetts Cooperative Fish and Wildlife Research Unit, University of Massachusetts, Amherst, Massachusetts, United States of America, **3** Massachusetts Division of Fisheries and Wildlife, Westborough, Massachusetts, United States of America, **4** Maine Department of Inland Fisheries and Wildlife, Bangor, Maine, United States of America

* glacial.erratic@yahoo.com

## Abstract

Freshwater mussels are critical to the health of freshwater systems, but their populations are declining dramatically throughout the world. The limited resources available for freshwater mussel conservation necessitates the geographic prioritization of conservation-related actions. However, lack of knowledge about freshwater mussel spatial distributions hinders decision making in this context. In this study, we assessed the distribution of twelve native freshwater mussel species across six Northeastern states (Connecticut, Rhode Island, Massachusetts, Vermont, New Hampshire, and Maine) in the United States using data collected from lentic and lotic environments by eight state agencies. We first modeled individual distributions using a maximum entropy (MaxEnt) model and then compiled distribution models to assess the distribution of freshwater mussel species richness. We also determined geographic prioritization for three conservation-related actions: species surveys, land protection, and population restoration of species of high conservation concern. We found that the percent of catchments predicted to have species occurrence (based on a probability threshold) varied across species, with *Elliptio complanata* (Eastern elliptio) predicted to occur in the greatest percent of available catchments (33.92%) and *Alasmidonta heterodon* (Dwarf wedgemussel) expected in the smallest percent (5.30%). The predicted overall species richness within our modeled catchments ranged from zero to all twelve species, with an average of two species per catchment. Although conservation priorities vary depending on the conservation action of interest, we found some areas of consistent importance including much of Maine and the southern reaches of the Connecticut River. An improved understanding of freshwater mussel distribution in a landscape framework will enable managers to implement more precise and efficient conservation interventions for these essential aquatic species.

**Data availability statement:** All relevant data for this study are publicly available from the USGS ScienceBase-Catalog repository (https://doi.org/10.5066/P13HZTMZ).

**Funding:** This research was supported by the following three grants: 1) U.S. Geological Survey Northeast Climate Adaptation Science Center G21AC10852. AR, GD. U.S. Geological Survey No role 2) Massachusetts State Hazard Mitigation and Climate Adaptation SHMCAP FY24 AR, GD Massachusetts Division of Fisheries and Wildlife https://www.mass.gov/orgs/division-of-fisheries-and-wildlife No role 3) ResilientMass FY25 JC, RQ Massachusetts Division of Fisheries and Wildlife https://www.mass.gov/orgs/division-of-fisheries-and-wildlife No role.

**Competing interests:** The authors have declared that no competing interests exist.

## Introduction

Increasing species loss [1], coupled with limited conservation resources [2], has heightened the imperative of thoughtful decision-making about natural resource use and conservation. There are a number of different approaches that have been taken towards spatial prioritization for various conservation actions, incorporating prioritization criteria such as evolutionary distinctiveness, biodiversity, resilience to global change and many others [3–5]. However, the appropriate weighting of each of these criteria can vary by species and location, and situation-specific prioritization approach is often necessary. Conservation that is targeted towards a particular taxonomic group and/or a limited geographic area can facilitate prioritization by clarifying the number and weighting of variables that must be considered, and it can necessitate a more tailored approach to ensure that findings are relevant [6].

One taxonomic group in need of conservation prioritization is the order Unionida, or freshwater mussels [7]. Freshwater mussels are found on every continent except Antarctica and are pillars of stream health, providing numerous ecosystem services including sediment stabilization, water filtration, nutrient cycling and storage, forage provisioning, and habitat creation [8,9]. They are also considered one of the most imperiled groups of aquatic organisms in the world [7,10,11] and are suffering from habitat alteration, invasive species, water pollution, and other challenges [12]. The threat to freshwater mussel populations is further enhanced by their natural history, which includes long generation times [13,14], specific habitat requirements [15], and a sedentary nature [15]. Additionally, freshwater mussels have a complex reproductive cycle whereby their larvae parasitize host fish, which they rely on for dispersal. As a result, impacts on host fish populations can also influence mussel populations [16].

Faced with the widespread and rapid decline of many freshwater mussel species, natural resource managers must decide how to focus conservation efforts to maximize the benefits of their actions. However, gaps in knowledge about where species are (or are likely to be based on environmental characteristics), has hindered decisions about where to focus land protection efforts [17,18]. Previous research into drivers of freshwater mussel distribution has identified several factors that influence species occurrence, including broad-scale factors like land cover, dispersal limitations, tidal influence, host fish occurrence, and dam density as well as reach-specific variables like stream discharge, stream bed composition, and shear stress (e.g., [19–22]). Species distribution models can build on this knowledge to identify areas of potential species occurrence and facilitate evidence-based decision making in the face of uncertainty [23].

For freshwater mussels, species distribution models are particularly well-suited in places like North America and Europe where there are sufficient occurrence records to provide the necessary data for modeling, but where knowledge gaps remain [24]. North America has the highest diversity of freshwater mussels in the world, but the International Union for Conservation of Nature (IUCN) lists roughly 45% of described freshwater mussel species in the continent as endangered, critically endangered, or extinct [25]. Throughout much of the United States (U.S.), freshwater mussel

conservation has generally consisted of state-by-state monitoring programs with uneven and uncoordinated survey efforts (e.g., [26–29]). Mussel distributions frequently traverse political boundaries, and this, coupled with the importance of the broader drainage to determining stream conditions, means a regional approach is necessary for effective management of these species. This is particularly true the northeastern United States which has smaller states than much of the rest of the country. In the last decade, two Northeastern U.S. regional working groups have emerged to coordinate mussel conservation in the area: one focused on *Alasmidonta varicosa* (Brook floater; [30]) and another on *Lampsilis cariosa* (Yellow lampmussel; [31]). While these groups foster information sharing and integration of effort across states, the efforts are limited to two species. A model of Unionida distribution in the region has not previously been published but would facilitate more informed regional management.

Using a predictive modeling approach, we delineated the distribution of 12 native freshwater mussels across six states in the Northeastern U.S. and described several possible means of geographic prioritization for freshwater mussel conservation. We first developed individual maximum entropy (MaxEnt) distribution models at the catchment scale using species presence-only data consolidated from state agencies. Based on the model outputs, we then calculated mussel species richness and the richness of species of conservation concern within catchments across the region. Finally, based on our modeled distributions and additional relevant variables, we assessed the priority of each catchment in the region for three conservation-related actions: species surveys, species population restoration, and land protection. Our results can help managers identify valuable catchments for supporting freshwater mussel conservation throughout the Northeast. More broadly, our approach can be applied to prioritizing various conservation actions for mussel species in other regions of the world that have adequate biological data.

## Materials and methods

### Study area

Our study area encompassed six states in the Northeastern U.S. including Maine (ME), New Hampshire (NH), Vermont (VT), Massachusetts (MA), Connecticut (CT), and Rhode Island (RI). There are extensive river networks and many freshwater lakes in the area. The largest lotic systems include the Merrimack, Connecticut, Kennebec, and Penobscot Rivers while the largest lake in the region is Lake Champlain, which lies on the Western border of Vermont. The area is comprised of three level III ecoregions (regions sharing ecosystem characteristics driven by biotic and abiotic traits) including the Northeastern coastal zone, Northeastern Highlands, and the Acadian plains and Hills [32]. The relatively high latitude of the Northeast, coupled with the Appalachian Mountain range, which runs through Northwestern Connecticut, Western Massachusetts, Vermont, and most of New Hampshire and Maine, keeps the average temperature of the region relatively cool compared to many other parts of the U.S. (average July/August stream temperature of 18.59°C; [33]). The area has large metropolitan areas, such as Boston and Providence, which are concentrated in the southern portion of the region and along the coast, but outside of these densely populated areas, much of the region is relatively rural and highly forested.

### Compiling freshwater mussel survey data

We compiled freshwater mussel species survey data from eight Northeastern state fish and wildlife agencies: Massachusetts Division of Fisheries & Wildlife, Connecticut Department of Energy & Environmental Protection, Vermont Department of Environmental Conservation, Vermont Fish & Wildlife Department, Rhode Island Department of Environmental Management, New Hampshire Department of Environmental Services, New Hampshire Fish and Game Department, and Maine Department of Inland Fisheries and Wildlife. We then filtered the data to records relevant to our study. We eliminated any records occurring prior to 1952, leaving us with records spanning between that date and 2022. Freshwater mussels are long-lived [14], so we chose to include all records from the last thirty years of data collection as the older records could still represent living individuals. We also eliminated eight mussel species (*Lasmigona compressa, Pyganodon grandis,*

*Potamilus capax, Lasmigona costata, Leptodea fragilis, Anodontoides ferussacianus, Potamilus alatus,* and *Alasmidonta raveneliana*) that were found exclusively in the Lake Champlain basin and provided limited presence locations for model creation. This left us with 12 native species for modeling (Table 1).

The majority of the data we consolidated did not include information about the survey method utilized to gather records and many did not include the type of specimen collected or the details necessary to calculate survey effort. For this reason, we did not filter or standardize our data based on these criteria. The data we consolidated contained multiple collection methods (e.g., snorkel surveys, view buckets), specimen types (e.g., live individuals, shells), and survey efforts. Although the diversity of approaches and specimens may have introduced some variability into our consolidated data [36], eliminating records without this information would have drastically reduced our sample size and impacted model fit. We reduced some of the effects of this variability by converting our data to presence only.

There was also variability in geographic precision and habitat type across our compiled data. Although some states and databases reported precise mussel locations, others (such as the NatureServe database; [37]) recorded more approximate locations represented by a survey line or polygon. To accommodate this variability, we consolidated all surveys to the catchment level. We included both lentic and lotic surveys in our data which represented 19.15% and 80.85% of the surveys respectively. We used only previously existing records of species occurrence, so no collection permits were required for this work.

## Species distribution models

**Geospatial framework.** We created individual distribution models (details described below) for each of the 12 species in our study using a maximum entropy (MaxEnt) modeling approach and the dismo package [38]. All analysis were performed in R 4.3.1 [39] and are available with the software release associated with this article [40]. All species were modeled at the catchment scale based on the NHDPlusV2 [41], which divides the U.S. into geographic units including watersheds, catchments, and 100-meter riparian buffers within the catchment or watershed. For the purposes of this paper, a catchment is defined as the area that drains directly to an NHD (National Hydrography Dataset) stream segment (excluding upstream contributions). A set of hydrologically connected catchments upstream of and flowing into a focal catchment is referred to as a watershed [41]. In addition to the NHDPlusV2 framework, we also used the U.S. Geological Survey (USGS)'s Hydrologic Unit Maps throughout the study [42]. The hydraulic unit framework breaks the United States

**Table 1. The twelve species of native freshwater mussel included in our study, their conservation status (based on the northeast Regional Species of Greatest Conservation Need, RSGCN, classification; [34]), and the total number of catchments with known occurrences of each species (Catchments based on [35]). There was a total of 68,191 catchments within our study area. Species that do not meet a certain level of conservation need are not listed on the RSGCN list.**

| Scientific name | Common name | RSGCN status | Number of catchments in our study area with known occurrences |
|---|---|---|---|
| *Alasmidonta heterodon* | Dwarf wedgemussel | Very high | 53 |
| *Alasmidonta undulata* | Triangle floater | Moderate | 780 |
| *Alasmidonta varicosa* | Brook floater | Very high | 209 |
| *Atlanticoncha ochracea* | Tidewater mucket | High | 124 |
| *Elliptio complanata* | Eastern elliptio | Not listed | 1754 |
| *Lampsilis cariosa* | Yellow lampmussel | High | 123 |
| *Lampsilis radiata* | Eastern lampmussel | Not listed | 484 |
| *Margaritifera margaritifera* | Eastern pearlshell | Moderate | 418 |
| *Pyganodon cataracta* | Eastern floater | Not listed | 844 |
| *Sagittunio nasutus* | Eastern pondmussel | High | 119 |
| *Strophitus undulatus* | Creeper | Not listed | 352 |
| *Utterbackiana implicata* | Alewife floater | Not listed | 210 |

into nested units that are assigned a two- to 12-digit code (the hydrologic unit code, or HUC) with larger HUC sizes representing smaller geographic units.

**Model covariates.** Our 12 species-specific MaxEnt models included a set of shared environmental layers including climate, geography, biota, and land use variables (Table 2). Our decisions regarding which environmental variables to include were based on literature review, expert feedback, and data availability. We chose to select variables individually for inclusion in the model rather than utilizing a suite of bioclimatic variables (e.g., those available from the dismo package; [38]), as this enabled us to select variables that more closely reflected the aquatic habitats of our study (e.g., using stream temperature in place of air temperature) and to more seamlessly integrate this project with related modeling efforts. However, many of the impacts of bioclimatic variables were nonetheless captured in our model. Most covariates were calculated at the catchment scale because we modeled the response variable at this spatial scale. However, for a few covariates that were less meaningful when averaged at the catchment level, we included the variable at an alternative scale (Table 2). Specifically, impervious cover was calculated within 100-m riparian buffers around National Hydrography Dataset (NHD) flowlines at the catchment scale; watershed area and percent wetland were determined through consideration of all catchments upstream of and flowing in to the focal catchment; and dam density and host fish variables were calculated at the HUC12 scale. After compiling a candidate list of variables (n = 20), we tested for relationships between them using Spearman's rank correlations ($\rho$). For any correlations exceeding $\rho > 0.70$, we consulted with experts to select the most ecologically relevant variables and eliminated others [43]. We then confirmed that all remaining variables had a variance inflation factor (VIF) below 5 using the vif function in the car package [44]. This process resulted in a total of 16 final environmental variables that were shared across all models (Table 2).

In addition to the shared environmental variables, our models also included 1–2 variables that were unique to the species being modeled. For each species, we included a layer that represented the recorded presence (or lack thereof) of the focal mussel species within the encompassing HUC8 watershed to account for historical distributions and limitations on dispersal. A primary aim of our model was to locate catchments of probable species occurrence that may not have been previously identified, so we took the approach of greater inclusion than exclusion when deciding how to approach modeling (i.e., we chose to include known species occurrence at the HUC8 rather than HUC10 scale as a covariate in our model). We also included a variable for host fishes for each freshwater mussel species. For host-specialist mussels (those whose host fish are restricted to members of a single family), we included the host fish species' probability of occurrence when available as an additional covariate (drawn from Rogers et al. *in review*). These values varied across focal species and were not tested for multicollinearity. For all mussels, we included the probability of any fish occurrence (Rogers et al., *in review*) to account for general host availability.

**Regularization parameter and feature class selection.** In addition to the model covariates, MaxEnt models have two primary parameter inputs: 1) the feature classes of the model covariates, which represent the type of mathematical transformation applied to model covariates and 2) the regularization parameter (also known as beta multiplier) which helps control model complexity and prevent overfitting. As a result of the regularization, environmental variable coefficients with insufficient contribution are greatly reduced or set at zero. By default, MaxEnt tests all available feature classes for the input variables, and we used this default setting for our model. We compared regularization parameters between 1 and 15 for each species using the ENMeval package [45] and proceeded with the regularization parameter that produced the highest area under the curve (AUC) value. This meant regularization parameters between 1 and 7 for our models. We chose to use AUC because we were interested in a consistent, threshold-independent metric and, despite its limitations [46], AUC proved to be the most appropriate measure of model quality given these constraints.

**Background point selection.** Sampling was not evenly distributed throughout our study region, and this, combined with our consolidation of species surveys to the catchment scale, necessitated a non-traditional approach to background point selection. First, to avoid selecting multiple background points from the same catchment (all of which had the same environment throughout), we reduced the available background points to the centroid of each catchment. MaxEnt models

**Table 2. Description of variables included in the species-specific maximum entropy (MaxEnt) models and their sources. Variables were drawn from a variety of sources and were chosen based on literature review, expert input, and data availability. Variable spatial scales (from McKay et al., 2012 and Hill et al., 2016) include: catchment, watershed, riparian buffer (~100 m on each side of the stream reach), and Hydrologic Unit Code 12 (HUC12).**

| Variable name | Spatial scale | Definition | Source |
|---|---|---|---|
| Tidal influence | Catchment | Presence or absence of a tidal influence | USGS, 2004 |
| Watershed area | Watershed | Watershed area (km²) | Hill et al., 2016 |
| Mean summer temperature | Catchment | Annual mean stream temperature (Jun 1st to August 31st) for the survey year (°C) | Letcher et al., 2016 |
| Dam density | HUC12 | The number of dams within the HUC12 divided by the area of the HUC 12 | State dam database |
| Winter floods | Catchment | Average number of daily flows between December 1 and March 31 that exceed the 95th percentile of daily flows across the entire year | USDA Forest Service Office of Sustainability and Climate, 2022 |
| Low flow date | Catchment | Average date of the center of the lowest 7-day flow of the year | USDA Forest Service Office of Sustainability and Climate, 2022 |
| Percent agriculture | Catchment | Percent of the catchment classified as pasture/hay land or row crop (averaged across 2001, 2004, 2006, 2008, 2011, 2013, 2016, and 2019) | Hill et al., 2016 |
| Percent wetland | Watershed | Percent of the catchment classified as woody or herbaceous wetland (averaged across 2001, 2004, 2006, 2008, 2011, 2013, 2016, and 2019) | Hill et al., 2016 |
| Percent impervious | Riparian buffer | Percent of each 30-m pixel classified as impervious anthropogenic materials (averaged across 2001, 2004, 2006, 2008, 2011, 2013, 2016, and 2019) | Hill et al., 2016 |
| Percent open water | Catchment | Percent catchment classified as open water (averaged across 2001, 2004, 2006, 2008, 2011, 2013, 2016, and 2019) | Hill et al., 2016 |
| Percent forest | Catchment | Percent deciduous, coniferous and/or mixed forest in the catchment (averaged across 2001, 2004, 2006, 2008, 2011, 2013, 2016, and 2019) | Hill et al., 2016 |
| Soil erodibility | Catchment | A relative index of susceptibility of bare, cultivated soil to particle detachment and transport by rainfall | Hill et al., 2016 |
| Slope | Catchment | Average slope | USGS, 2004 |
| Waterbody volume | Catchment | Average lentic system volume | Hill et al., 2016 |
| Probability of any fish species occurrence | HUC12 | Average summed probability of occurrence of any fish species | Rogers et al. *in review* |
| Occurrence within encompassing HUC8* | HUC8 | Known record of focal mussel species occurrence within the encompassing HUC8 based on our dataset. | Project dataset |
| Preferred host occurrence* | HUC12 | Average summed proportional abundance of preferred host fish. Available for the following mussels and their host fish:<br>*Margaritifera margaritifera*: *Salmo trutta, Oncorhynchus mykiss, Salmo salar, Salvelinus fontinalis*<br>*Sagittunio nasutus*: *Perca flavescens, Micropterus nigricans, Lepomis macrochirus, Lepomis gibbosus,* and *Lepomis auritus*<br>*Lampsilis cariosa*: *Morone americana* and *Perca flavescens*<br>*Atlanticoncha ochracea*: *Morone americana* | Rogers, unpublished analysis |

*Variables that differed across models.

assume that all points within the study area have an equal probability of being field sampled [47], but our survey sites were unevenly distributed. To account for this, we eliminated any catchment that was within a HUC10 that had no survey records for any species from selection as a background point. HUC10 was chosen to avoid excessive exclusion of catchments while ensuring that catchments well outside the sampled range were eliminated. This reduced the likelihood that our model reflected probability of sampling rather than probability of occurrence. We biased the probability of being selected as a background point to reflect the probability of the encompassing HUC10 being sampled. Finally, we further reduced the available background points for each species individually by eliminating catchments with known focal species

occurrences. The inclusion of background points with known occurrences in models is possible in most MaxEnt models [48]. However, consolidating the available background points to catchment level meant we greatly restricted the number of available background points, and a very large number of known presences were being incorporated as background.

We selected 10,000 background points from the final pool of available points for each species, which corresponded to an average of 18.16% of the available points (range = 14.47–100%). We used a weighted random sampling approach to select these points with the probability of a point being sampled as background reflecting its probability of being surveyed (calculated as the percent of all surveys that occurred in each catchment). An exception to this approach was taken for *Elliptio complanata* (Eastern elliptio). This species occurs so widely throughout the region that there were only 9,338 background points available for selection, and all available points were used as background for this species' model.

**Model evaluation and visualization.** There is no current consensus regarding the preferred metric to evaluate MaxEnt models [46,49–51]. We used a combination of the AUC [49], which despite its limitations, is the most commonly used metric for MaxEnt evaluation, and the Continuous Boyce Index (CBI; [52,53]), which is appropriate for presence only data and in some instances outperforms AUC [54]. We considered models with AUC values greater than 0.7 to be acceptable. Additionally, to describe model accuracy, we determined the number of false negatives (i.e., catchments that the model did not predict to have a focal species but that had records of occurrence), and we report the number of catchments that had probable species occurrence according to individual thresholds (based on maximum sensitivity and specificity, described below) but that did not have any documented occurrences of the focal species.

Using R package ggplot2 [55], we created distribution maps illustrating the probability of occurrence for each species. After modeling each species, we tested for correlations between each species pair in the predicted probability of occurrence across catchments using Spearman's rank correlations (ρ) and the cor.test function in R's stats package [39].

## Calculating species richness

In addition to assessing the probability of each species' occurrence across the Northeast individually, we also assessed overall mussel species richness and the richness of mussel species of conservation concern at the catchment level. For overall species richness, we summed the total number of species predicted to occur in a catchment. For species of conservation concern, we determined the number of species listed as Northeastern Regional Species of Greatest Conservation Need (RSGCN) which ranks species as very high, high, or moderate risk in each catchment (Table 1; [34]). We defined individual thresholds to consider a species present or absent from a catchment as the value that maximized the sum of specificity and sensitivity [23,55–57] using the evaluate function in the dismo package and considering background points as pseudoabsences.

We tested for similarities in the overall species richness per catchment and the RSGCN species richness per catchment using Spearman's rank correlation (ρ) and the cor.test function in R's stats package. Lastly, we also calculated the average number of species per catchment across states.

## Catchment prioritization

We also prioritized catchments across the Northeast for three conservation-related actions including species surveys, land protection, and species population restoration. For the purposes of this study, we use population restoration to mean both adding individuals to areas where their species has been extirpated and adding individuals to supplement existing populations (IUCNSSC, 2013). We chose these three activities from a large number of possible conservation-related actions because they were, according to experts, common actions within agency control. Calculations of catchment priority for each conservation action relied on a combination of our modeled species distribution output and data drawn from external sources (Table 3). Although there are many variables and goals that may contribute to decision making for each of these actions, we focused on broadly applicable environmental variables that could serve as a foundation upon which managers could layer additional criteria (described below). We ultimately produced maps illustrating the conservation value of each

**Table 3. Summary table of the variables used to calculate catchment prioritization for survey efforts, population restoration, and protection. "+" indicates variables that we added to the priority score calculation, while "-" indicates variables that we subtracted to the priority score calculation. For example, species survey priority was calculated by the following equation: standardized species richness – binary prior survey occurrence. HUC = hydrologic unit code. We limited our analysis of catchment priority for population restoration to species with a Regional Species of Greatest Conservation Need (RSGCN) rank of high or very high conservation concern.**

| Variable | Spatial scale | Source | Catchment prioritization | | |
|---|---|---|---|---|---|
| | | | Species surveys | Population restoration | Land protection |
| Prior surveys | Catchment | Raw survey data | – | | |
| Overall species richness | Catchment | Model projection | + | | + |
| Recorded species occurrence | HUC8 | Raw survey data | | + | |
| Percent protected land | Catchment | USGS, 2024 | | + | |
| Projected land use change | Catchment | Clark Labs, 2021 | | – | |
| Percent forest cover | Catchment | Hill et al., 2016 | | + | + |
| Percent wetland | Catchment | Hill et al., 2016 | | | + |
| Climate change resilience | HUC12 | Rogers unpublished analysis | | | + |

catchment using ggplot2 [55], ranked catchment priority based on each criterion, and compared conservation priorities across states in the region.

Although we provide prioritization guidance at the catchment scale, many management decisions may be made at smaller or larger spatial scales. After identifying a catchment of priority, managers may choose to further optimize how they utilize their resources within the catchment to achieve conservation goals. Alternatively, managers may choose to aggregate catchments in an encompassing HUC12 or larger unit to facilitate decision making. Average priority values and maximum species richness at a HUC12 scale are available in the data release associated with this article [58].

**Species surveys.** To prioritize catchments for species surveys, we used a combination of whether the catchment had previously been surveyed and its predicted species richness. We considered catchments with high predicted richness and no prior surveys to be higher priority for this conservation action. We scaled species richness to be between zero and one and created a binary score indicating whether each catchment had or had not been surveyed after 1992. We included only records from the last 30 years of data to enhance the relevancy of our results. We then subtracted the survey binary value from the scaled richness and scaled the result to be between zero and one. Higher values indicated higher priority, and lower values indicated lower priority for survey effort.

**Population restoration.** We limited our analysis of catchment priority for population restoration to species with an RSGCN rank of high or very high conservation concern (Table 1). A key component of successful population restoration is that the habitat is of sufficiently high quality to support the reintroduced species [59], so we incorporated indicators of habitat quality available at the regional scale into our assessments. Catchments were considered of greater population restoration priority if they had higher percent protected land, higher percent forest cover within the watershed, and lower percent projected land use change. We calculated percent protected land from the Protected Areas Database (PAD; [60]) and included all designations (national monuments, mitigation lands, conservation easements, etc.) in our analysis. For our measures of projected land use change, we used a GIS layer of predicted land use change between 2018 and 2050 [61] and restricted the data to increased cropland or increased urbanization. We then calculated the percent of each catchment that was expected to experience one of those two land use changes. Percent forest cover was drawn from the USGS National Land Cover Database (NLCD) for 2019 [62].

As with other prioritization calculations, we scaled and added or subtracted each variable (Table 3) and then scaled the result to be between zero and one. Population restoration efforts occur within the species' historical geographic range [59], so any catchments within a HUC8 watershed that did not contain any previous records were considered ineligible for

population restoration and not given a priority score. Higher values indicate higher priority, and lower values indicate lower priority for species population restoration.

**Land protection.** Catchment prioritization for land protection was based on maximizing the quality of habitat and the number of species protected in both the current and in predicted future climates. To this end, we considered catchments to be of higher priority for land protection if they had greater overall species richness, higher percent forest cover within the watershed, higher percent wetland within the watershed, and a higher likelihood of species composition stability in the face of climate change (hereafter "climate resilience"). Percent forest cover and percent wetland were both drawn from the 2019 USGS NLCD [62]. The likelihood of a catchment having stable species populations in the face of climate change was based on unpublished analysis by co-author Rogers, who modeled the probability of occurrence within HUC12s for each of the 12 mussel species under a future scenario of +3°C. Each variable (species richness, percent forest cover within the watershed, percent wetland within the watershed, and climate resilience) was scaled and summed. The resulting value was then scaled to be between zero and one to generate the land protection prioritization value. More positive values indicate higher priority, and lower values indicate lower priority for land protection. To determine the extent to which current protected land matched priority score, we also calculated the correlation between percent protected land and current percent protected land using cor. test in R's stats package.

## Results

### Species distributions

Our final dataset included 12 freshwater mussel species with recorded occurrences in 53–1,754 catchments (mean = 456: Table 1) out of the 66,658 total catchments across the Northeast. All states except Rhode Island had at least one HUC8 that contained five or fewer surveys (n = 4,4,2,2, and 3 for CT, ME, NH, RI, and VT respectively; Fig 1).

Each of the 12 freshwater mussel species-specific distribution models had AUC scores greater than 0.95 (Table 4) and had CBI values greater than 0.96 indicating good fit. The percent of catchments predicted to have species occurrence based on the maximum sensitivity and specificity probability threshold varied across species with *Elliptio complanata* (Eastern elliptio) predicted to occur in the greatest percent of available catchments (35.47%) and *Alasmidonta heterodon* (Dwarf wedgemussel) expected in the smallest percent (3.70%; Fig 2, Table 4).

The predicted probability of occurrence for each of the 12 native freshwater mussel species we modeled in the Northeastern United States. The probability of occurrence was calculated based on species-specific maximum entropy (MaxEnt). (U.S. states reprinted from U.S. Census Bureau [64])

Our models largely corroborated known freshwater mussel occurrences, and they identified catchments where species were likely to be but where there were not known occurrences. Across all 12 species and 68,191 catchments, there were 596 instances of catchments that had recorded species occurrence, but that our models predicted would not be present based on the max sensitivity/specificity threshold (i.e., probable false negatives). Conversely, there were 8,920 instances of catchments that our models predicted were likely to have occurrences but that did not have species records (Table 4). The species with the greatest number of catchments with probable species occurrence based on the model but no documented occurrence records was *Alasmidonta undulata* (Triangle floater; n = 1226), while *A. heterodon* had the fewest (n = 29). The species with the greatest number of probable false negatives was *E. complanata* (n = 151), and the species with the fewest was *A. heterodon* (n = 2).

Most of the species were moderately correlated with one another in their predicted distribution throughout the region (Average ρ = 0.46 ± 0.15; Fig 3). *Sagittunio nasutus* (Eastern pondmussel) and *Margaritifera margaritifera* (eastern pearlshell) showed the lowest correlation in predicted probability of occurrence (ρ = 0.81, p < 0.001), while *E. complanata* had high correlations with *A. undulata* (ρ = 0.81, p < 0.001). For further information about how the covariates influenced distribution and for maps displaying binary predicted presence/absence (based on thresholds used in species richness

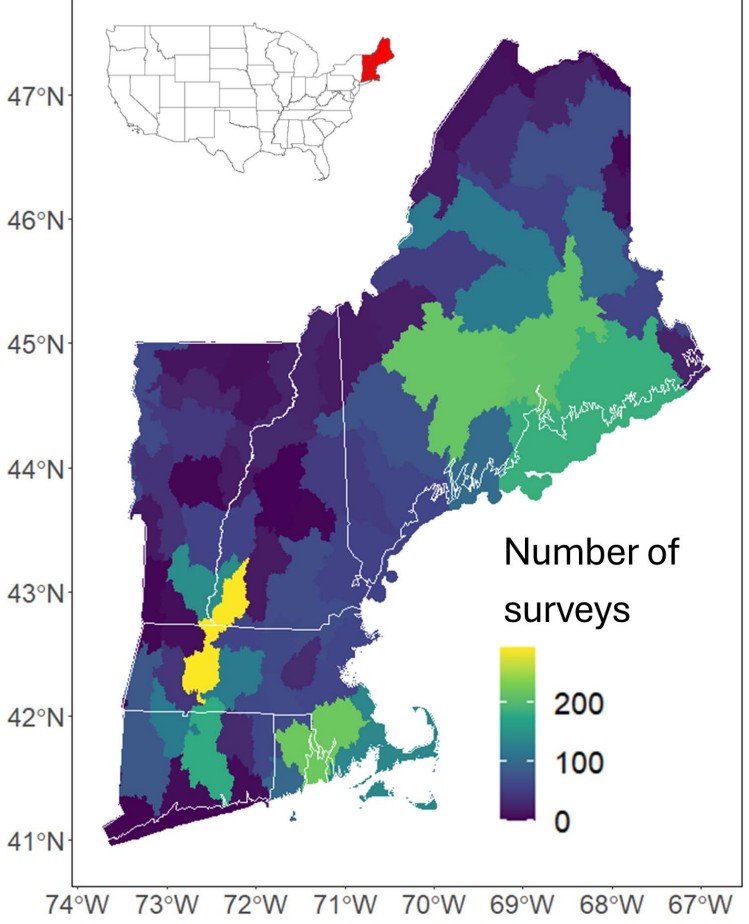

**Fig 1. Study area and number of surveys per HUC8 watershed for all species combined.** Our study area included (from bottom left to top right) Connecticut, Rhode Island, Massachusetts, Vermont, New Hampshire, and Maine which are the six northeastern-most states in the continental United States (top left inset). Known occurrence of the focal mussel species within the encompassing HUC8 was included as a layer in our models to account for dispersal, but some HUC8s had very few survey records. (U.S. states reprinted from U.S. Census Bureau, 2023).

calculations), see the data release associated with this article [58]. Maps displaying the standard deviation of model estimates based on k-fold cross validation (k = 10) are also available in the data release [58].

## Species richness

The predicted overall species richness ranged from zero to all twelve species, with an average of two species per catchment (Fig 4a). Rhode Island and Maine had the greatest average predicted species richness per catchment while New Hampshire had the lowest (Table 5). All catchments with more than 10 predicted species were located in the Connecticut River watershed, driven in part by *A. heterodon* which occurs exclusively within this watershed. A prioritized list of catchments based on species richness and each of the following conservation-related actions is available in the data release associated with this article [58].

The number of RSGCN species ranged from zero to all seven species (Fig 4b), but the distribution was highly skewed with a mean of 0.78 RSGCN species per catchment. There was a strong correlation between the distribution of RSGCN species and overall species richness ($\rho = 0.84$, $p < 0.001$).

**Table 4. Summary statistics for each individual species model including beta multipliers (i.e., regularization parameter), areas under the curve (AUC), continuous boyce index (CBI), and maximum summed sensitivity and specificity values for each species (used as the threshold for binning a species as present or absent from each catchment). Percent possible habitat is the percent of our study area that had a probability of occurrence greater than zero. The number of false negatives indicates the number of catchments where a species was not expected to occur based on the model but where there were records of occurrence, and the number of possible new catchments indicates the number of catchments where a species did not have a recorded occurrence, but where the model predicted a species to occur.**

| Scientific name | AUC | CBI | Maximum summed sensitivity and specificity value | Percent possible habitat | Number of false negative catchments | Number of possible new catchments with species |
|---|---|---|---|---|---|---|
| *Alasmidonta heterodon* | 0.99 | 0.97 | 0.06 | 3.70 | 2 | 29 |
| *Alasmidonta undulata* | 0.95 | 0.99 | 0.24 | 18.72 | 134 | 1107 |
| *Alasmidonta varicosa* | 0.99 | 0.98 | 0.10 | 10.97 | 13 | 729 |
| *Atlanticoncha ochracea* | 0.99 | 0.99 | 0.14 | 7.83 | 4 | 447 |
| *Elliptio complanata* | 0.96 | 0.99 | 0.29 | 35.47 | 151 | 769 |
| *Lampsilis cariosa* | 0.99 | 0.98 | 0.12 | 11.90 | 2 | 767 |
| *Lampsilis radiata* | 0.97 | 0.99 | 0.20 | 20.42 | 34 | 944 |
| *Margaritifera margaritifera* | 0.97 | 0.99 | 0.33 | 16.86 | 25 | 990 |
| *Pyganodon cataracta* | 0.94 | 0.99 | 0.25 | 31.52 | 141 | 1226 |
| *Sagittunio nasutus* | 0.99 | 0.99 | 0.20 | 6.22 | 32 | 98 |
| *Strophitus undulatus* | 0.98 | 0.99 | 0.18 | 15.48 | 39 | 932 |
| *Utterbackiana implicata* | 0.99 | 0.96 | 0.20 | 15.70 | 19 | 882 |

## Catchment prioritization

*Species surveys* – Catchments showing high survey priority were concentrated in Maine and the southern reaches of the Connecticut River (Fig 5a). Overall, Maine showed the highest average state-wide survey priority score, while Vermont had the lowest, although all states were all quite similar (Table 5).

*Species population restorations* – Catchment priority for species population restoration was determined by the same variables for all modeled species, but candidate areas for species population restoration were restricted by individual species' recorded occurrence (i.e., only catchments within a HUC8 with a recorded occurrence were eligible). The result was that there was a single underlying priority map for the entire region, but different species had different parts of the region available for population restoration (Fig 5b). As such, the number of catchments available for each species' population restoration varied across species, with *A. heterodon* having the fewest (n = 11,659) and *A. varicosa* the most (n = 47,504) available catchments. There was also variation in average catchment priority score for each species' population restoration area. *S. nasutus* had the lowest average catchment priority score (mean = 0.54, SE = 0.001), while *A. heterodon* had the highest (mean = 0.61SE = 0.002). *A. varicosa* had an average score of 0.59 (SE = 0.001), *L. cariosa* had an average of 0.60 (SE = 0.001), and *Atlanticoncha ochracea* (tidewater mucket) had an average of 0.57 (SE = 0.001).

*Land protection* – High priority catchments for land protection were relatively evenly distributed throughout the region with some marked cold spots, often near major cities (e.g., Boston; Fig 5c). Maine, the state with the highest average percent protected land per catchment, also had the greatest average protection priority score (mean = 0.58, SE = 0.001). However, there was very little correlation between protection priority score and current percent protected land in either Maine ($\rho$ = 0.05, p < 0.001) or the region as a whole ($\rho$ = 0.08, p < 0.001) suggesting that high priority areas for protection for mussels are not always the areas that are protected. Rhode Island had the lowest average land protection priority score (mean = 0.46, SE = 0.004).

## Discussion

Our results document the distribution of native freshwater mussels in the Northeastern U.S. and provide guidance to managers on how they might consider prioritizing catchments for a variety of different conservation-related actions. Although

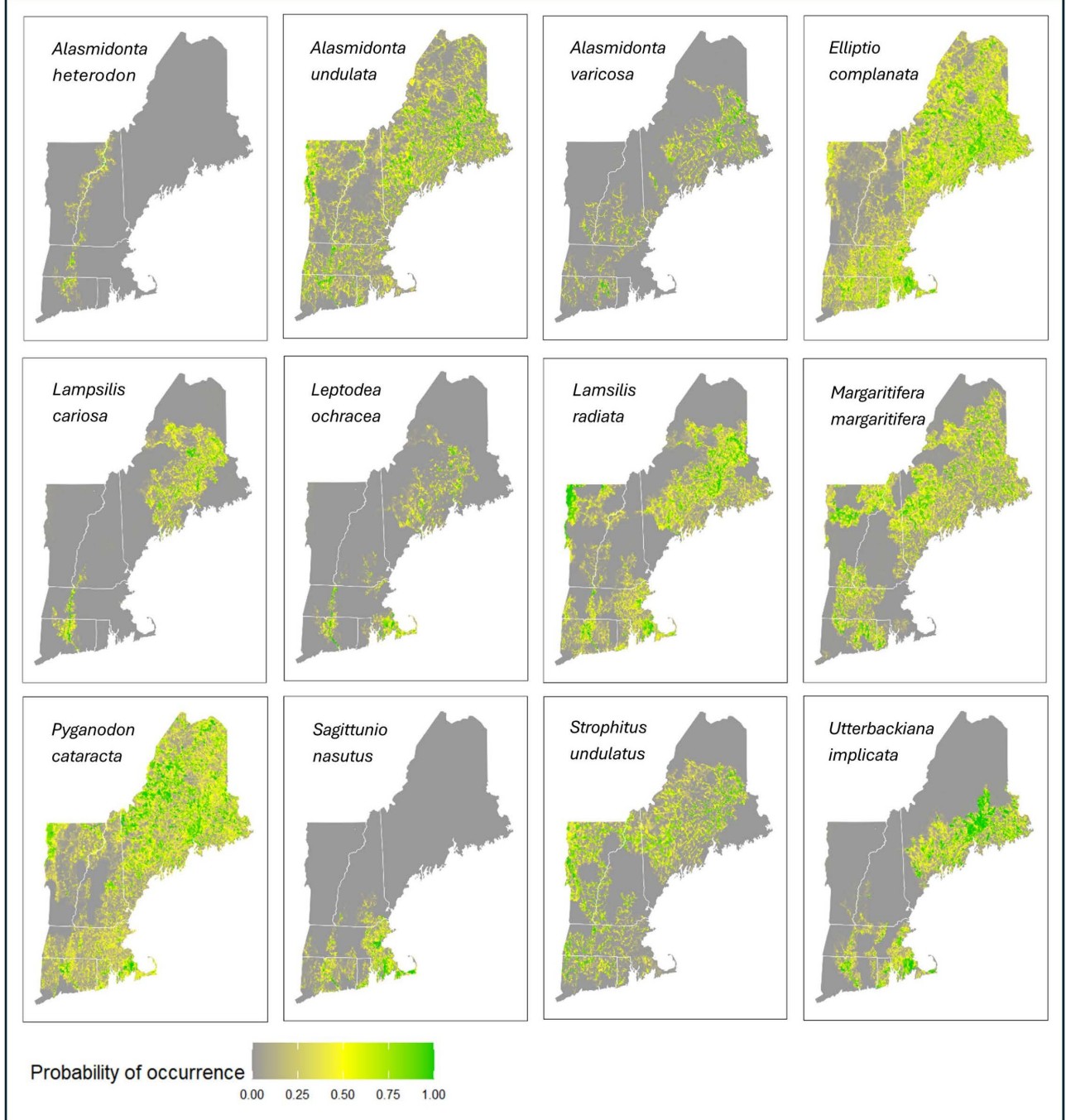

**Fig 2. Predicted probability of occurrence.**

there has been considerable effort invested in spatial conservation prioritization based on large numbers of species and extensive landscapes [63], this effort adds to the smaller body of work focused on conservation prioritization for a limited number of species in a smaller spatial extent which can provide more targeted conservation guidance [65]. Additionally, we simultaneously considered several different conservation actions rather than focusing on one action alone, which is a strategy that has been applied in other species (e.g., [66]) to support holistic guides for managers.

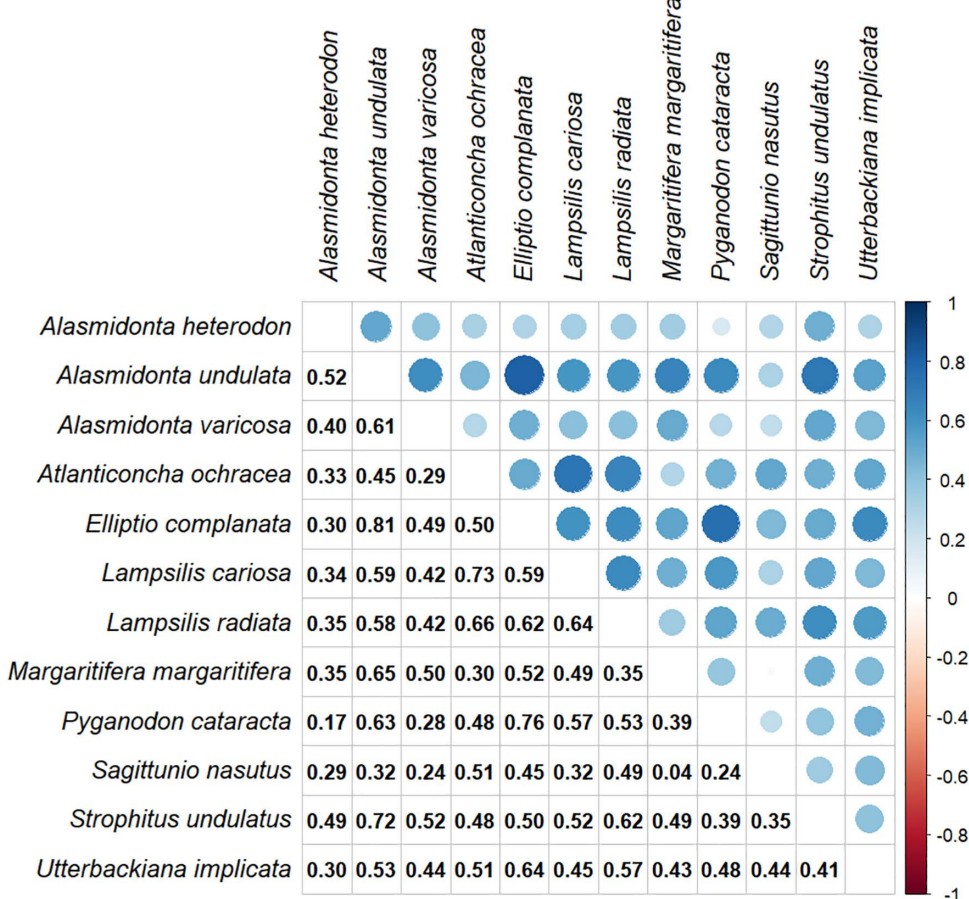

**Fig 3. Correlation matrix of the predicted probability of occurrence between each species pair.** The upper right half of the figure visually represents the strength of the correlation through the size and color of the circle, with larger darker circles representing strong positive correlations and smaller redder circles representing strong negative correlations. The lower left half reports the spearman's correlation coefficient value.

There have previously been state-by-state efforts in the Northeastern U.S. to map freshwater mussel species distributions based on known occurrences (e.g., [26–29]. By working at a regional scale, we were able to capture broad scale spatial patterns of freshwater mussel species occurrence that may facilitate interstate coordination while also maintaining sufficient spatial granularity to keep our results useful to managers working at the state or catchment level. Our modeling approach also enabled us to move beyond known occurrences and identify areas and states with potentially high or low species richness. The highest priority catchments in our study varied depending on the focal species and the conservation action of interest, but there were some areas that emerged as particularly important for freshwater mussel conservation across analyses, including much of Maine and the southern reaches of the Connecticut River.

At the individual species level, our models confirmed state distribution maps and identified catchments where habitat was suitable for species occurrence, but where they had not yet been recorded. In some cases, the model identified areas of occurrence that extended well beyond what state experts currently consider to be likely habitat, indicating a limitation in our model. For example, *L. cariosa*'s predicted range follows the Penobscot River watershed to Maine's northwestern border, but state experts (co-author Swartz) consider the species unlikely to occur in this upper-most portion of the watershed. To further the models' relevance, managers can hone our predicted distributions based on local knowledge of distribution limitations that we may have been unable to incorporate as well as reach-scale habitat characteristics that were not available at a regional scale (e.g., substratum composition; [67]).

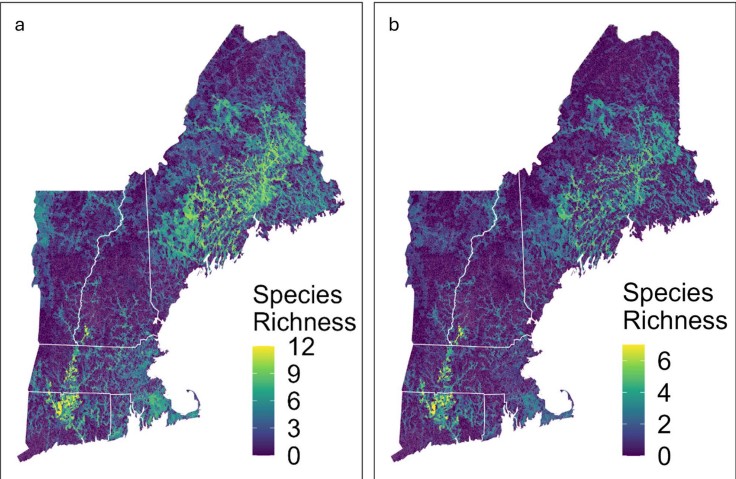

**Fig 4. Species richness.** (a) overall species richness (of 12 native mussel species) and (b) species richness of Regional Species of Greatest Conservation Need (RSGCN; up to 7 total species) based on individual species' predicted probability of occurrence. Catchments with more than 5 RSGCN species or more than 10 overall species were found exclusively in the Connecticut River Basin. (U.S. states reprinted from U.S. Census Bureau [64]).

**Table 5. The total number of mussel surveys, average predicted species richness per catchment, and average (standard error in parentheses) priority value for species population restoration, species surveys, and land protection. Priority scores are unitless and were scaled to be between 0 and 1. Not all species were predicted to occur in all states, resulting in N/A (not applicable) priority scores.**

| State | Number of surveys | Average predicted species richness (#/ catchment) | Average population restoration priority score | | | | | Average survey priority score | Average land protection score |
|---|---|---|---|---|---|---|---|---|---|
| | | | *Atlanticoncha ochracea* | *Alasmidonta varicosa* | *Lampsilis cariosa* | *Alasmidonta heterodon* | *Sagittunio nasutus* | | |
| Connecticut | 196 | 1.44 | 0.56 (0.004) | 0.58 (0.002) | 0.56 (0.004) | 0.56 (0.004) | 0.57 (0.002) | 0.55 (0.001) | 0.45 (0.002) |
| Maine | 1267 | 2.53 | 0.60 (0.001) | 0.59 (0.001) | 0.61 (0.001) | 0.80 (0.044) | N/A | 0.58 (0.001) | 0.58 (0.001) |
| Massachusetts | 773 | 2.05 | 0.48 (0.002) | 0.59 (0.002) | 0.59 (0.005) | 0.59 (0.005) | 0.49 (0.002) | 0.56 (0.001) | 0.52 (0.001) |
| New Hampshire | 157 | 1.06 | 0.60 (0.003) | 0.61 (0.002) | 0.68 (0.005) | 0.66 (0.003) | 0.59 (0.003) | 0.54 (0.001) | 0.54 (0.001) |
| Rhode Island | 138 | 2.30 | 0.49 (0.005) | 0.66 (0.009) | N/A | N/A | 0.52 (0.004) | 0.54 (0.004) | 0.46 (0.004) |
| Vermont | 295 | 1.34 | 0.59 (0.017) | 0.63 (0.003) | 0.59 (0.017) | 0.63 (0.003) | 0.59 (0.017) | 0.54 (0.001) | 0.53 (0.001) |

In addition to identifying individual species distributions, the models also were useful in determining probable areas of freshwater mussel species co-occurrence. We found a strong correlation in predicted distributions across most freshwater mussel species. This is consistent with previous studies which have found that many of our modeled species tend to co-occur in diverse assemblages [68]. However, biases in survey effort may mean that co-occurrences are more likely to be identified than not (i.e., surveys often target known mussel beds). We were also able to assess overall freshwater mussel richness, and we found that richness was high in Maine and the southern reaches of the Connecticut River, and low in Vermont and New Hampshire. However, caution should be used when interpreting the relatively low richness in these latter two states because both New Hampshire and Vermont had few surveys across large percentages of their land

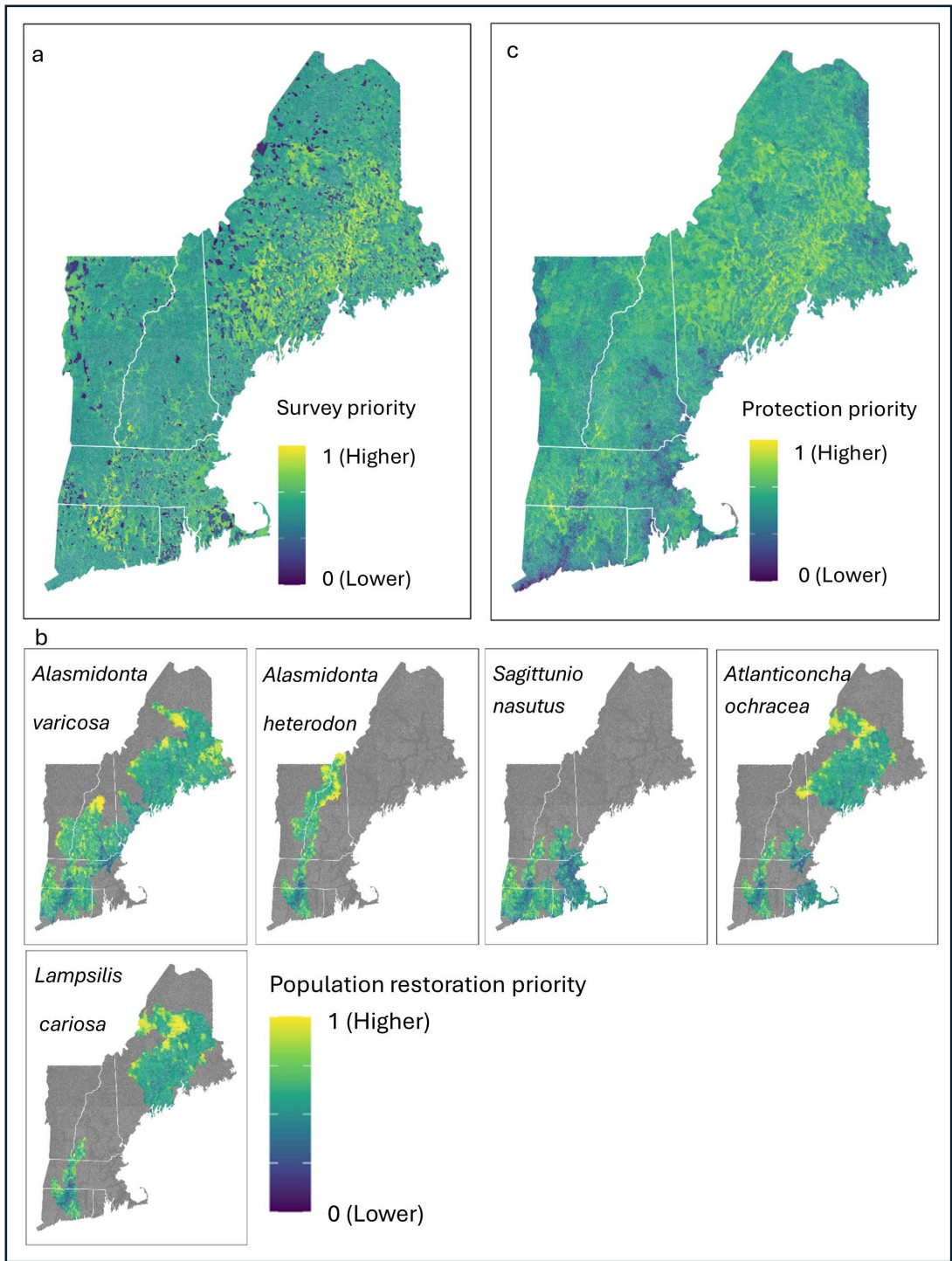

**Fig 5. Catchment priority.** Priority scores for (a) surveys, (b) land protection, and (c) population restoration. Survey and protection priority were assessed for all species together, while we assessed population restoration priority individually for species that were considered high or very high Regional Species of Greatest Conservation Need (RSGCN) priority.

mass, particularly in the central regions of the states and in southwest Vermont. This low survey number meant that there were fewer opportunities to detect a species occurrence within each HUC8 which was an important layer in our model (see data release for covariate permutation importance). Additionally, many of the species found exclusively in the Lake Champlain basin in Vermont were eliminated from our study due to their restricted range, artificially reducing the apparent species richness in this area. Total species richness and the richness of only species of conservation concern were highly correlated, which suggests that managing for species of regional conservation need may also benefit other freshwater mussel species across the Northeast.

Despite ongoing efforts at the state level to survey for freshwater mussels, there are still large parts of the Northeast that have not been surveyed. Some of these under-surveyed areas are likely to contain viable mussel populations. For example, the Connecticut River Watershed, which boasts some of the of highest predicted species richness in our study region, was less than 10% surveyed as of 2008 [69], and these numbers remain low. Methods for relatively rapid assessment have been developed for wadeable streams [70], but surveying is still a resource-intensive process. Adding to the expense, most states in our study area rely on contractors which are costly and there are limited contractors of sufficient expertise (co-author Carmignani, personal observation). For these reasons, a strategic approach is necessary to maximize the benefit of survey efforts. Our calculation of survey priority showed areas of high priority in Maine and the southern reaches of the Connecticut River, largely mirroring where the model predicted highest species richness. Prior surveys had a relatively limited impact on future survey priority because few catchments had previous effort. Given this limitation, it may be valuable to combine our results with expert-based assessments (e.g., [71]) to further clarify priority catchments.

Species population restoration is increasingly considered as a management tool in areas where freshwater mussel populations have been greatly reduced or extirpated [7,19]. In the Northeast, population restoration has not yet been implemented, but it is being considered for *A. varicosa* in Connecticut and Massachusetts [72] and potentially other species. We found that species population restoration priority was lowest surrounding the urban areas of the Northeast (i.e., Boston, Massachusetts; New Haven and Hartford, Connecticut) and that the largest assemblages of high priority catchments were concentrated in the northern extents of the region, largely following the Appalachian Mountain Range. While all species had high priority catchments within their range, *S. nasutus*, whose range is restricted to the southern reaches of the region, did not have the same large collection of high priority catchments as were seen for other species. It is worth noting that, because we did not have species abundance data, we could not model variation in abundance across the region. For this reason, areas that are identified as high priority for population restoration may in fact not need this management approach.

Habitat suitability is only the first of several steps in determining population restoration suitability [73], and managers hoping to undertake these actions will need to incorporate a number of additional considerations such as disease risk, invasive species pressure, social-ecological feasibility, and whether the cause of the species decline has been addressed [72,74]. Decisions surrounding the significance of these factors may vary by agency, species, and location and factors that predict occurrence in one area may not always do so in another [75] so a nuanced, fine-scale approach to incorporating these variables could be beneficial.

Land protection is perhaps the most commonly considered conservation action in prioritization schemes [76]. Land protection is valuable for freshwater mussels as land use can significantly influence water quality and, in turn, mussel species richness [77]. Land protection can take a variety of forms including land acquisition for conservation [78] or conservation agreements on private land [79–81]. Protection through acquisition is limited by land availability, and knowing which catchments are of higher or lower priority can help managers quickly determine whether it is worth investing resources in acquiring land as it comes available. Based on our analysis, high priority protection catchments were concentrated in Maine, which is also currently the state with the highest average percent land protection per catchment. However, we found a very low correlation between the current land protection and land protection priority in Maine and throughout the

region (Maine: $\rho = 0.04$ region: $\rho = 0.08$), suggesting that there is ample opportunity for additional land protection. Our models provide guidance on where these new land protections would best be implemented if available.

Conservation-related actions are often made through consideration of multiple species, geographic ranges, and time points. Although a focused assessment of priority can improve how well the effort reflects the needs of a specific taxonomic group, the value of our prioritization results could be elevated by next situating them in a broader context. For example, combining our results with findings from a broader scale, such as the entire eastern U.S. or the Nearctic ecoregion [24], could better capture species status and landscape-scale patterns of distribution. Similarly, combining our results with models of other taxonomic groups (e.g., turtles, macroinvertebrates) could provide more inclusive but refined lists of priority catchments for conservation that would benefit a diversity of organisms. We did not include invasive species in our model, and although *Dreissena polymorpha* (zebra mussel) and *Dreissena rostriformis* (quagga) are not of great concern in our study area, *Corbicula fluminea* (Asian clam) was identified in the region in 1990 [82] and is a growing concern that may influence native mussel distribution even more heavily going forward. Future research could incorporate the impact of this invasive competitor in driving native mussel distribution. Our models also only considered the current distribution of freshwater mussels across the Northeast (with the exception of our protected area prioritization), but consideration of climate change will be critical to effective long-term conservation planning. Many of our species can be expected to experience range shifts as a result of changing climate suitability (Rogers, unpublished analysis). Nonetheless, by considering freshwater mussel distribution alone, and incorporating their distribution into a broader context, managers can more effectively and efficiently conserve these unique, and critically important animals.

## Acknowledgments

We are grateful to Melissa Doperalski (New Hampshire Fish and Game Department), Laura Saucier (Connecticut Department of Energy and Environmental Protection), Michelle Graziosi (Vermont Department of Environmental Conservation), and Corey Pelletier (Rhode Island Department of Environmental Management) who shared their state's freshwater mussel survey records and provided biological expertise and valuable feedback on modeling decisions. Additional mussel survey records were provided by the Massachusetts Division of Fisheries and Wildlife, Maine Department of Inland Fisheries and Wildlife, New Hampshire Department of Environmental Services, and Vermont Fish and Wildlife Department. Gabriel Fournier, Gab DeVito, and Tyler Pelt assisted in the digitizing, extracting, and tidying of raw data. Any use of trade, firm, or product names is for descriptive purposes only and does not imply endorsement by the U.S. Government.

## Author contributions

**Conceptualization:** Rebecca O'Brien, Graziella V. DiRenzo, Allison H. Roy, Jason Carmignani, Rebecca M. Quiñones.

**Data curation:** Rebecca O'Brien, Jennifer B. Rogers.

**Formal analysis:** Rebecca O'Brien.

**Funding acquisition:** Graziella V. DiRenzo, Allison H. Roy, Jason Carmignani, Rebecca M. Quiñones.

**Investigation:** Rebecca O'Brien.

**Methodology:** Rebecca O'Brien, Graziella V. DiRenzo, Allison H. Roy, Jason Carmignani, Rebecca M. Quiñones.

**Project administration:** Graziella V. DiRenzo, Allison H. Roy, Rebecca M Quiñones.

**Resources:** Graziella V. DiRenzo, Jason Carmignani.

**Software:** Rebecca O'Brien.

**Supervision:** Allison H. Roy, Rebecca M. Quiñones.

**Validation:** Rebecca O'Brien, Graziella V. DiRenzo, Allison H. Roy, Jason Carmignani, Rebecca M. Quiñones, Beth I. Swartz.

**Visualization:** Rebecca O'Brien.

**Writing – original draft:** Rebecca O'Brien.

**Writing – review & editing:** Rebecca O'Brien, Graziella V. DiRenzo, Allison H. Roy, Jason Carmignani, Rebecca M. Quiñones, Jennifer B. Rogers, Beth I. Swartz.

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
