## [Decision Letter · Decision Letter 0]

25 Feb 2025

PONE-D-25-03714Native freshwater mussel species distribution and catchment prioritization for mussel conservation across the Northeastern United States

PLOS ONE

Dear Dr. O'Brien,

Thank you for submitting your manuscript to PLOS ONE. After careful consideration, we feel that it has merit but does not fully meet PLOS ONE’s publication criteria as it currently stands. <small>Therefore, we invite you to submit a revised version of the manuscript that addresses the points raised during the review process.</small>

<small>Please submit your revised manuscript by Apr 11 2025 11:59PM. If you will need more time than this to complete your revisions, please reply to this message or contact the journal office at plosone@plos.org . </small>

We look forward to receiving your revised manuscript.

Kind regards,

Sanja Puljas

Academic Editor

PLOS ONE

Journal Requirements:

4. Please note that funding information should not appear in any section or other areas of your manuscript. We will only publish funding information present in the Funding Statement section of the online submission form. Please remove any funding-related text from the manuscript.

“This research was supported by the following three grants:

1) U.S. Geological Survey Northeast Climate Adaptation Science Center

G21AC10852. 

AR, GD. 

U.S. Geological Survey

No role

2) Massachusetts State Hazard Mitigation and Climate Adaptation 

SHMCAP FY24

AR, GD

Massachusetts Division of Fisheries and Wildlife

https://www.mass.gov/orgs/division-of-fisheries-and-wildlife

No role

3) ResilientMass

FY25

JC, RQ

Massachusetts Division of Fisheries and Wildlife

https://www.mass.gov/orgs/division-of-fisheries-and-wildlife

No role”

6. We note that you have indicated that there are restrictions to data sharing for this study. PLOS only allows data to be available upon request if there are legal or ethical restrictions on sharing data publicly. For more information on unacceptable data access restrictions, please see http://journals.plos.org/plosone/s/data-availability#loc-unacceptable-data-access-restrictions.

7. We note that Figures 1,2,4 and 5 in your submission contain map/satellite images which may be copyrighted. All PLOS content is published under the Creative Commons Attribution License (CC BY 4.0), which means that the manuscript, images, and Supporting Information files will be freely available online, and any third party is permitted to access, download, copy, distribute, and use these materials in any way, even commercially, with proper attribution. For these reasons, we cannot publish previously copyrighted maps or satellite images created using proprietary data, such as Google software (Google Maps, Street View, and Earth). For more information, see our copyright guidelines: http://journals.plos.org/plosone/s/licenses-and-copyright.

 a. You may seek permission from the original copyright holder of Figures 1,2,4 and 5 to publish the content specifically under the CC BY 4.0 license. 

Additional Editor Comments:

The MS adds valuable new information to the literature and the content of the MS should be published. While the general appearance is acceptable, the MS needs to be revised before final acceptance. The manuscript was evaluated by two reviewers who expressed concerns with which I agree. In general, this study should be placed in an international context, as the research in this form has a regional character.

Please see the comments at the end of this letter for the reasons why I have come to this decision.

Reviewers' comments:

Reviewer's Responses to Questions

**Comments to the Author**

1. Is the manuscript technically sound, and do the data support the conclusions?

Reviewer #1: Partly

Reviewer #2: Yes

2. Has the statistical analysis been performed appropriately and rigorously? 

Reviewer #1: Yes

Reviewer #2: Yes

3. Have the authors made all data underlying the findings in their manuscript fully available?

Reviewer #1: Yes

Reviewer #2: No

4. Is the manuscript presented in an intelligible fashion and written in standard English?

Reviewer #1: Yes

Reviewer #2: Yes

5. Review Comments to the Author

Reviewer #1: Thank you for your manuscript. I found it to be an interesting read and fits well in the journal. However, in order to proceed, I would like to see some more applicability in the international context. At the moment, this is a regional study from the north-eastern United States. While the science is sound and the manuscript reads well, you need to incorporate aspects in the introduction and discussion that would warrant its inclusion in the international journal. Otherwise, consider a regional journal such as Freshwater Mollusk Biology and Conservation or the American Malcological Bulletin.

Reviewer #2: The paper “Native freshwater mussel species distribution and catchment prioritization for mussel conservation across the Northeastern United States” is a relevant contribution to this area of knowledge and indeed may have an interest to a broad audience other than those interested in freshwater mussels. In fact I would suggest modifying the title to reflect this (as it is the title may lead to narrowing down the range of readers), something like “Catchment prioritization for freshwater mussel conservation across the Northeastern United States using distribution modelling tools”. The manuscript is very well written and easy to follow. The conclusions are sound and based on the data presented.

My main concerns deal with the methods section. Although it is already quite extensive and the approaches are generally sound, it often fails to explain clearly what was done or how was it done. Here are some examples:

1) Having used data from since the 1950’, was the potential effect of outdated presence data been taken into account? If so, how? If not, this potential effect must be discussed further ahead. Some variables used have changed dramatically over the course of this time, so if they are used as explanatory the data will suffer from this shortcoming.

2) Have the variables been used in the models as equivalent in importance and not weighted? From some sections of the text, it seems that there was some kind of variable weighting at some point, but this is not explained.

3) Was there no attempt to use bioclimatic variables? Why not? This is somewhat surprising as these may be used to predict future distributions, and this should be taken into account when prioritizing conservation actions.

4) Why were invasive species not used as potential explanatory variables?

5) Methods for validating the models must be better accounted for in the methodology.

6) Calculations for estimating prioritizations are difficult to replicate with the provided explanations alone. Maybe presenting an equation or example would help.

The discussion is very clear, yet I feel that it would benefit from contracting the work with broader geographical scales. The interest to national-level conservation is briefly mentioned and should be extended (comparing with other species examples for instance). Comparison with other countries would also be relevant, especially considering there are quite a few published papers modelling freshwater mussel distribution. The impact of climate change should also be discussed, especially considering that no modelling of future distributions was made.

There are also some minor concerns through the manuscript, where clarification of concepts or simplification of language must be reviewed. Some examples:

1) Replace “Study Geography” by “Study Area”

2) “Ample freshwater” (line 120) is a dubious expression. Better use “extensive river network”, large water bodies, etc.

3) “level III Ecoregions” are referring to the EPA classification I presume. International reader will in general not understand what this refers to, so must be explained.

4) “These calculations” (line 278). Which calculations? Surely the ones to estimate catchment prioritization, but no calculations were presented at this point.

5) Reintroductions: Although well explained, it would be better referring these concepts to the ones presented in the IUCN guidelines to reintroductions.

6) Table 5: Presents state letters instead of the actual names, without explaining in the legend. These are not broadly understood, so either write the names in full in the table or explain in the legend.

(The whole manuscript must be revised carefully for these minor issues)

Finally, the short title does not seem short at all. Suggest: “Conservation prioritizing of freshwater mussels in Northeastern US”

6. PLOS authors have the option to publish the peer review history of their article (what does this mean? ). If published, this will include your full peer review and any attached files.

**Do you want your identity to be public for this peer review?** For information about this choice, including consent withdrawal, please see our Privacy Policy .

Reviewer #1: **Yes: ** Michael W. Klunzinger

Reviewer #2: No

---

## [Author Response · Author response to Decision Letter 1]

17 Apr 2025

Thank you to the reviewers for their careful review of our manuscript. The suggested edits have greatly improved the paper, and we hope we have addressed them satisfactorily. This paper is published through the USGS, so in addition to the revisions recommended by the reviewers, this draft also includes revisions in response to a USGS reviewer and the Bureau Approving Official. Below, I have included a table with individual responses to each reviewer’s comments as well as the USGS reviewer’s comments and responses. In the cover letter for this manuscript, there is a table detailing our responses to each reviewer's comment. Individual responses to reviewer comments can be found in the reviewer response document included with the resubmission package.

---

## [Editor Report · Decision Letter 1]

24 Apr 2025

Catchment prioritization for freshwater mussel conservation in the Northeastern United States based on distribution modelling

PONE-D-25-03714R1

Dear Dr. O'Brien,

We’re pleased to inform you that your manuscript has been judged scientifically suitable for publication and will be formally accepted for publication once it meets all outstanding technical requirements.

Kind regards,

Sanja Puljas

Academic Editor

PLOS ONE
---

## [Editor Report · Acceptance letter]

PONE-D-25-03714R1

PLOS ONE

Dear Dr. O'Brien,

I'm pleased to inform you that your manuscript has been deemed suitable for publication in PLOS ONE. Congratulations! Your manuscript is now being handed over to our production team.

Kind regards,

on behalf of

Dr. Sanja Puljas

Academic Editor

PLOS ONE